

# Variation in ant-mediated seed dispersal along elevation gradients

Israel Del Toro and Relena R. Ribbons

Biology Department, Lawrence University, Appleton, WI, United States

## ABSTRACT

Ant-mediated seed dispersal, also known as myrmecochory, is a widespread and important mutualism that structures both plant and ant communities. However, the extent to which ant functional types (e.g., granivorous generalists vs. myrmecochorous ants) across environmental gradients affect seed removal rates is not fully understood. We used a replicated, standardized seed removal experiment along elevation gradients in four mountain ranges in the southwestern United States to test predictions that: (1) seed removal rates would be greater at lower elevations, and (2) seed species identity influences seed removal rates (i.e., seeds from their native elevation range would be removed at higher rates than seeds outside of their native elevation range). Both predictions were supported. Seed removal rates were ∼25% higher at lower elevation sites than at higher elevation sites. The low elevation *Datura* and high elevation *Iris* were removed at higher rates in their respective native ranges. We attribute observed differences in dispersal rates to changes in ant community composition, functional diversity, and abundance. We also suggest that temperature variation along the elevation gradient may explain these differences in seed removal rates.

## INTRODUCTION

Myrmecochory—ant-mediated seed dispersal—is an ecologically important and ubiquitous mutualistic interaction that exists between ants and plants (*Ness & Bressmer, 2005*), occurring in over 11,000 species, and across 70 plant families (*Lengyel et al., 2010*). Seeds of myrmecochorous plants contain lipid-rich appendages called elaiosomes, which serve as nutritional rewards for ants (*Warren & Giladi, 2014*; *Gómez & Espadaler, 2013*; *Ness & Bressmer, 2005*). Variation in seed removal rates can be explained by ant morphological traits (*Fokuhl, Heinze & Poschlod, 2012*) and seed traits like elaiosome and plant size (*Leal, Leal & Andersen, 2015*; *Peters, Oberrath & Bohning-Gaese, 2003*). Ants also influence the dispersal capacity of a seed by making seeds unavailable to other potential seed predators (*Leal, Leal & Andersen, 2015*) and also disperse non-elaiosome bearing seeds through seed harvesting or granivory (*Christianini et al., 2007*; *Taber, 1999*). In both cases ant-mediated seed dispersal plays a key role in shaping vegetation community structure (*Del Toro, Ribbons & Pelini, 2012*; *Ghobadi et al., 2015*), as ants may serve as seed dispersers or predators depending on the context.

Corresponding author
Israel Del Toro, israedt@gmail.com, Israel.deltoro@lawrence.edu

While ant-mediated seed dispersal or predation plays an important role in vegetation communities, this role likely varies across elevation and habitat gradients. "Sky islands" are individual mountains isolated from larger continuous mountain ranges by vastly different habitat types, in this case the Chihuahuan and Sonoran deserts at the lower elevations. The sky islands of the southwestern United States provide an excellent system for studying how myrmecochory varies along a continuous elevation and habitat gradient. This gradient extends from desert scrub to subalpine coniferous forest (*Brusca & Moore, 2013*), which is an excellent natural experiment as both plant and arthropod communities tend to change systematically along gradients (*Andersen, 1997*; *Del Toro, Ribbons & Pelini, 2012*).

Ant-mediated seed dispersal shapes vegetative communities (*Rissing, 1986*) and this important mutualism (or commensalism see *Warren et al., 2018*) is potentially resilient to increased temperatures in temperate forests in the eastern U.S. (*Stuble et al., 2014*). However, less is known about the potential breakdown of ant-mediated seed dispersal in a desert to forest mountain gradient. We suspect strong connections between myrmecochorous plants and ants within desert communities (*Leal, Leal & Andersen, 2015*). This could be due to their extreme climates and rapidly changing environmental gradients (for example, intense deluges of rain and dramatic shifts in temperatures in a single day (see *Andersen, Del Toro & Parr, 2015*). The intense competition for resources in extreme environments may also mean some ants are reliant on extrafloral nectaries and other plant resources to persist (*Aranda-Rickert, Diez & Marazzi, 2014*). Stress for fundamental resources has also led to strengthened ant-plant mutualisms in water-stressed environments (*Pringle et al., 2013*). Thus, we predict that myrmecochory may play a strong role in shaping vegetation communities in desert environments. We also expect tightly coupled ant-plant mutualisms that are species-specific for plants that differ among habitat types and elevations. For example, ant species and plant seed traits that have evolved in the same environment are more likely to be associated with one another, than those that have not.

We documented seed removal rates, and the effect of seed species identity on seed removal rates along this gradient. Given previous research at our study locations, we knew ant community composition differed among mountains in the Sky Islands (*Andersen, 1997*), with higher ant abundances observed at lower elevations on the mountains. We predicted that: (1) rates of seed removal would be higher at lower elevations, and (2) individual rates of seed removal for each seed species would be highest at a seeds' native elevation range (the elevation range at which each plant species is endemic, and is observed growing in the wild).

## MATERIALS AND METHODS

### Study site and organisms

We conducted this study on four sky islands in the southwestern United States of America in June and July 2015 (see File S1, Table 1). At each elevation band we deployed 10 seed depots, each containing 100 seeds, with 25 seeds from four different plant species (Table 1). To test whether seed removal rates were dependent on species identity, we selected seeds from plants native to different habitats along the elevation gradient. We built seed depots

**Table 1  Plant study species details and observed seed-dispersing ant genera.**

| Plant common name | Plant scientific name | Native elevation range (m.a.s.l.) | Observed seed-dispersers (ant genera) |
|---|---|---|---|
| Sacred Datura | *Datura wrightii* | 1,200–1,800 | *Pogonomyrmex, Novomessor* |
| Littleleaf Sumac | *Rhus microphylla* | 1,800–2,400 | *Pogonomyrmex, Novomessor, Pheidole* |
| Rocky Mountain Iris | *Iris missouriensis* | 2,400–3,200 | *Formica, Myrmica* |
| Common Oat | *Avena sativa* | Middle east (non-native control) | *Pogonomyrmex, Novomessor, Liometopum* |

using laminated white index cards covered with an overturned, reinforced, disposable plastic plate (Dixie Co. Atlanta GA, USA) held in place with lawn staples (Easy Gardener Inc. Waco TX, USA) to prevent seeds from being blown by wind or being removed by larger granivores. Ants were allowed to enter and exit the depot through 1 cm openings, cut out around the covering plate. We counted the remaining seeds every 12 h over a 48-hour period. Each depot was deployed and counted within hours of each other, within a single mountain site.

We selected seeds from four different plants (Datura, Iris, Oat, and Sumac) to test our second prediction, that seeds would be removed at greater rates in their native elevation range. *Datura wrightii* is the only species with an elaiosome-bearing seed and is common in desert scrub habitat (typically found at elevations <1,800 m.a.s.l.) (*Carter et al., 1997*). Based on our field observations Datura seeds were typically removed by Harvester ants (*Pogonomyrmex* spp). Little Leaf Sumac (*Rhus microphylla*) fruits were observed in the field being harvested by various ant genera (*Pogonomyrmex, Aphaenogaster* and *Pheidole*) at mid elevations from 1,800 m.a.s.l. to 2,400 m.a.s.l. The Rocky Mountain Iris (*Iris missouriensis*) normally occurs only in habitat above 2,400 m.a.s.l. (*Brusca & Moore, 2013*), and ants in the genera *Formica* and *Myrmica* were observed actively removing seeds from the plants. We harvested seeds from Datura, Iris, and Sumac plants within one week of deploying the experimental seed depots to account for differences in phenological emergence. Lastly, we used an oat seed (*Avena sativa*) that served as our control group, as it is not native to the study region (thus lacks any eco-evolutionary dynamics with native ant communities) nor does it contain an elaiosome, but may be a source of nutrients for opportunistic and granivorous ants.

To assess ant community composition and abundances, we used a 16 pitfall trap quadrant array at each sampling location. Pitfall traps were deployed for the same 48 h when seed removal rates were observed, at each elevation band and site and were directly adjacent with the location of the seed depots. We report ant incidence as the number of pitfalls that captured an individual genus, which is a more conservative metric of ant communities.

## Statistical analyses

We used a generalized linear mixed model (GLMM) to test for an additive (non-interacting predictor variables) effect between elevation and seed species, and to test our first prediction that seed removal would vary across elevation (m.a.s.l.) and site (latitude and longitude coordinates/mountain name). We treated time, elevation, and seed identity as fixed effects,

and specified a Poisson distribution for the count data (the number of seeds removed from seed depots). The total number of seeds removed over time was compared among seed species and elevation bands using repeated-measures analysis of variance (RMANOVA). To test our second prediction (that seed identity influences seed removal across elevations) we used a GLMM and Chi square tests for each species separately, and included site as a random factor.

We conducted an overdispersion test, modified from Ben Bolker (overdisp_fun; see Supplementary Material lines 24–32), to test for equal variance in GLMMs given a particular distribution family (Poisson in our case). This function was applied to all glmer models and results are reported in the supplementary materials. Overdispersion was not detected in the global model M1 (which tested all seeds among all elevations and sites), but was present in the individual models of Iris and Sumac. For Iris and Sumac, overdispersed models were fitted using a quasi-Poisson distribution, which allowed us to estimate which sites and elevations are driving the primary detected patterns. For Iris- the significant difference in high elevations is driven by one site, MOG, which indicates there is no overall pattern. For Sumac–variable responses at high elevation sites (2,800 m) may be dampening any possible trends, which indicates there is no overall pattern.

See the supporting information file (File S1 for the detailed code and dataset used in the analyses. All analyses including GLMMs were implemented in R statistical program version 3.2.3 (*R Development Core Team, 2016*) using the "lme4" package (*Bates et al., 2015*) and Chi square tests in the "car" package (*Fox & Weisberg, 2011*). To identify differences in ant community composition we used Principal Component Analysis (PCA) implemented in the "FactoMineR" package and visually inspected PCA biplots.

## RESULTS

Consistent with our first prediction, seed removal rates were higher at low elevations (1,600 and 2,200 m; Fig. 1). Total seed removal rates were similar among sites at 1,600 m and those at 2,200 m, but seed removal rates were 23.8% lower at 2,800 m than at the lower elevations (Fig. 1, File S1), a pattern consistent with our first prediction. Our repeated measures analyses suggested a strong additive effect between elevation and seed species (Table 2, $p < 0.001$). In all analyses (global model and seed specific models), time influenced the number of seeds removed. This suggests that the longer the depots are active, the more seeds are removed, ranging from 25% to 58% over the 48-hour time period.

Datura and Oat seeds accounted for most of the seed removal (Figs. 2A, 2C, 13% and 18% respectively), whereas Iris and Sumac seeds were removed at lower rates (2B, 2D, ~5% for either species). Datura seed removal was highest at the low elevation sites, with no differences between the mid and high elevation sites. Oat seed removal tended to be greater at mid elevations and approximately the same at low and high elevations (Fig. 2C), compared to the other seed species. Iris and Sumac seeds were removed less frequently from our depots, with Iris seeds having greater removal rates (Fig. 2B) while Sumac had the lowest removal rates between these two species (Fig. 2D), at high elevation sites after 48 h. Our repeated measures analyses suggest that elevation as a constraint on ant and

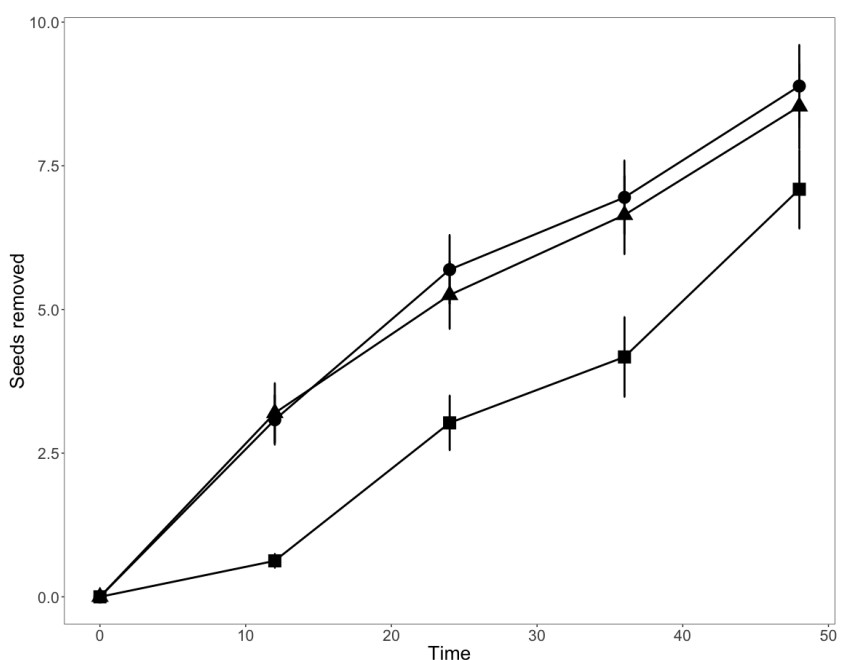

**Figure 1   Mean seed removal rates per bait station along the elevation gradient over a 48-hour period.**
Squares, 1,600 m.a.s.l., circles, 2,200 m.a.s.l. and triangles, 2,800 m.a.s.l. Bars indicate standard error about
means.

**Table 2   RMANOVA table of total seeds removed over time, partitioned by each seed species using Type II Wald Chi sq. tests.**

|  | Fixed effects | Chi sq | Pr (<Chi sq) |
|---|---|---|---|
| Global model | Hours | 690.722 | <0.001 |
|  | Elevation: seed | 560.986 | <0.001 |
| *Datura wrightii* | Hours | 300.565 | <0.001 |
|  | Elevation | 27.533 | <0.001 |
|  | Site | 9.304 | 0.026 |
| *Iris missouriensis* | Hours | 258.427 | <0.001 |
|  | Elevation | 13.511 | 0.001 |
|  | Site | 9.343 | 0.025 |
| *Avena sativa* | Hours | 411.720 | <0.001 |
|  | Elevation | 12.230 | 0.002 |
|  | Site | 25.450 | <0.001 |
| *Rhus microphylla* | Hours | 76.433 | <0.001 |
|  | Elevation | 8.117 | 0.017 |
|  | Site | 4.902 | 0.179 |

plant communities, was an important factor in explaining seed removal rates for all four
species, and site level effects were important for Datura and Oats, but Iris or Sumac.

We examined ant communities at the generic level using pitfall trap data and principal
components analysis, and found a distinct ant community at high elevations that is

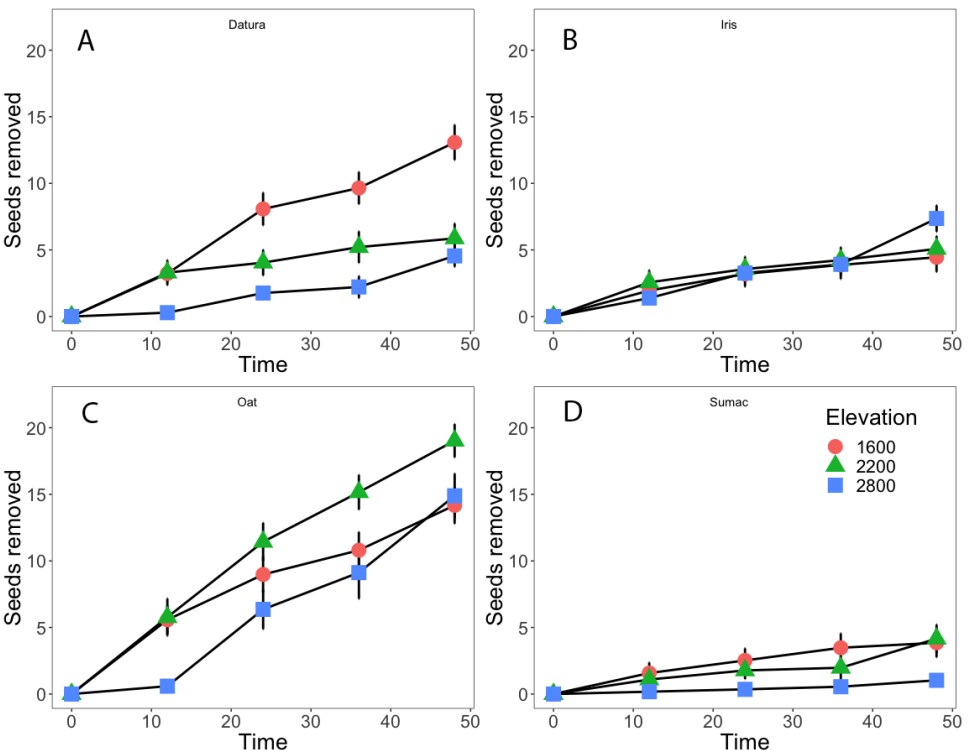

**Figure 2  Seed removal through time.** Seeds removed at three elevation classifications (low elevation circles, mid elevations green triangles, and high elevations blue squares) over a 48 hour period and for four seed types: (A) Datura (B) Iris (C) Oat and (D) Sumac. Bars indicate standard error.

characterized by *Myrmica* and *Lasius* spp. (Fig. 3). We found mid elevation ant communities are characterized by more *Tapinoma, Camponotus, Aphaenogaster* and *Liometopum* spp. than either the high or low elevation communities (Fig. 3). Low elevation communities are characterized by *Dorymyrmex, Crematogaster, Pogonomyrmex, Tetramorium*, and *Myrmecocystus* spp. (Fig. 3).

# DISCUSSION

Our findings highlight that while seed removal by ants may be widespread across environmental gradients, it may be more pronounced in arid low-elevation compared with mesic high-elevation habitats for ant communities. Observed seed removal rates were greater in lower elevations than in higher elevations. Seed removal was strongly dependent on seed species identity and its plant species native elevation range, which was consistent with our predictions. The pattern of total seed removal may be attributable to higher abundance of seed dispersing ants, especially Harvester ants (*Pogonomyrmex* spp., *Novomessor* spp.). These two species of harvester ants are more common at lower elevations (*Taber, 1999*) and have multiple traits which make these ants ideal dispersers of seeds that are dropped while foraging and not consumed (*Warren & Giladi, 2014*; *Zelikova, Dunn & Sanders, 2008*), including a specialized dietary preference for seeds (*Taber, 1999*). These

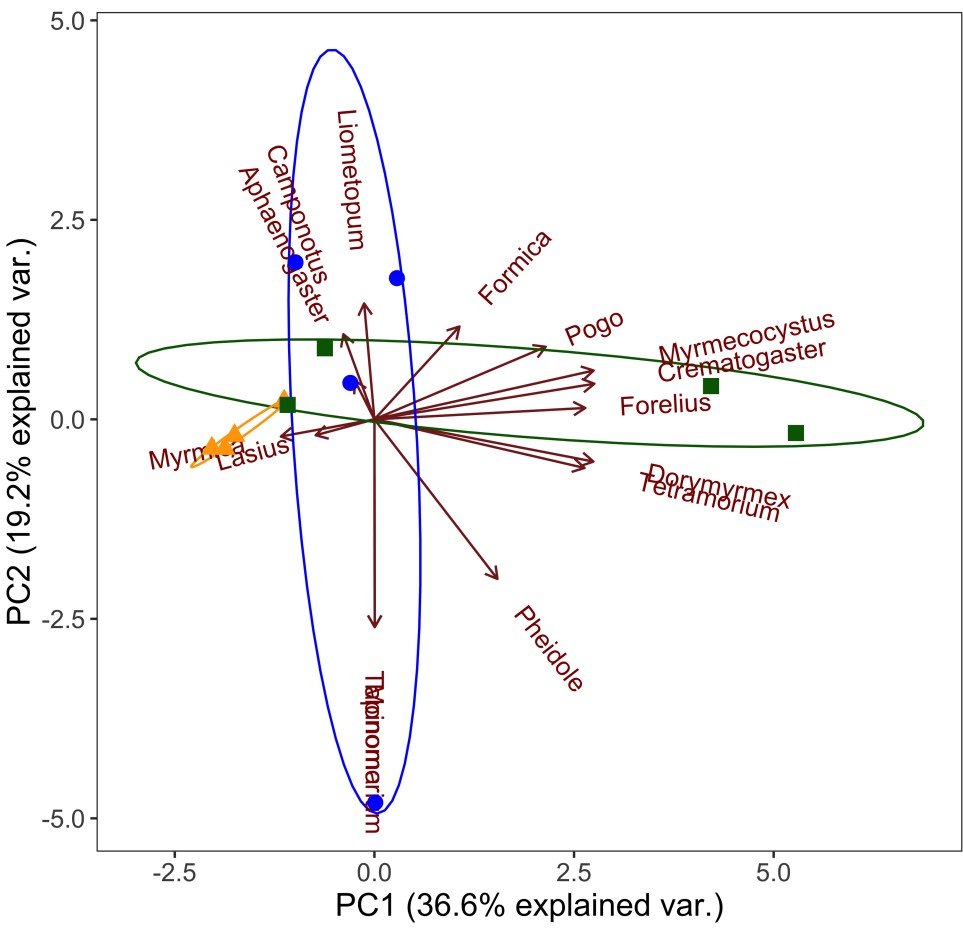

**Figure 3  Principal components analysis of ant community composition.** Grouped by elevation band (triangles- high, circles- mid, squares- low) at the generic level with ellipses indicating 95% confidence intervals for each elevation band (yellow- high, blue- mid, green- low).

harvester ants were observed removing seeds of Datura and Sumac, as well as the control Oat, but never dispersing the Iris.

In other temperate region studies, ant abundance is correlated with unimodal species richness and abundance patterns with low ant diversity/abundance at high elevations, with a peak of diversity/abundance in mid to lower elevations (*Bharti et al., 2013*; *Lessard et al., 2011*). Our results suggest a similar pattern of ant abundance could be driving seed removal rates, as both seed removal and ant abundances were higher at low elevation sites. Myrmecochory may be more prevalent in ecosystems where high ant abundances can be advantageous for dispersing seeds, (e.g., Datura in low-elevations). The mutualistic relationship of harvester ants (*Pogonomyrmex*) and *Datura wrightti* has been carefully documented and the effects of Datura seed diets on ant reproductive output have been experimentally tested (*Marussic, 2006*). Ant community composition also changes along these elevation gradients, with the lower elevations having high abundances of hot-climate specialist and generalized Myrmicinae species, and high elevation sites tend to have lower

abundances of ants which were mainly cold climate specialist and opportunistic species (*Andersen, 1997*) (File S1). The observed ant community differences may explain the patterns of seed removal along this gradient. Behaviorally dominant and abundant genera (beyond harvester ants) may remove more seeds at low elevations, while opportunistic genera may remove fewer seeds at high elevation sites. Higher rates of seed removal at lower elevations may also correlate with higher temperatures, which leads to increased ant activity levels (*Stuble et al., 2014*).

Seed species identity also influenced observed seed removal patterns. Datura and Iris seeds were removed at higher rates in their native elevation ranges (1,600 m and 2,800 m, respectively). The pattern detected in our study suggests that at low elevations, ant communities have highly active and abundant, seed-dispersing species (File S1). Mid-elevation sites tend to have preferentially granivorous ant communities (as indicated by the higher removal of oat seeds at mid-elevations). High-elevation sites have relatively lower ant abundances (File S1) largely comprised of opportunistic species. Our results may reflect an unequal distribution of functional ant diversity along these elevation gradients. This uneven distribution of functional diversity places concern on seed-dispersing ant species that may be sensitive to climate change, especially in temperate (high elevation) ecosystems (*Del Toro, Silva & Ellison, 2015*) or instances where climate-driven ecological mismatches between seed drop and ant activity occur (*Warren & Giladi, 2014*).

Future work should explore the specifics of the network of ants interacting with various seed species being dispersed along environmental gradients. For example, by tracking the individual ant species removing individual seeds, we could identify major seed dispersing ant species and the total influence they have on structuring vegetation communities. We recognize the potential for the influence of seed-drop phenology influencing the seeds being dispersed, a pattern that was documented in eastern North American forests (*Warren & Giladi, 2014*). This may partially explain low rates of Sumac removal, since it tends to drop its seeds earlier in summer than the other seeds (*Carter et al., 1997*). Given that *Datura wrightii* is a species of conservation interest and a myrmecoschorous plant with an elaiosome, we also suggest a replicated experiment across a finer resolution elevation gradient would be useful for predicting the fate of *Datura* in the future.

## CONCLUSIONS

Although myrmecochory is widespread and important in structuring plant and animal communities (*Warren & Giladi, 2014*; *Del Toro, Ribbons & Pelini, 2012*), this relationship is not equally distributed along elevation and habitat gradients. Furthermore, a single ant species can perform a majority of seed dispersal, such as *Pogonomyrmex* spp. and *Novomessor* spp. in this study or *Aphaenogaster rudis* in the Great Smoky Mountains (*Zelikova, Dunn & Sanders, 2008*). We highlight that for some species (Datura and Iris) their ant-mediated dispersal rate is highest in their native elevation range, which may suggest strong mutualistic links between ants and these plant species. The work on seed dispersal along elevation gradients allows us to explore how key ant-mediated ecosystem processes respond to environmental cues and help us predict how communities might respond to future climatic and habitat change.

## ACKNOWLEDGEMENTS

We thank Christian Rodriguez and Bill McKay for data collection and genera identification assistance, Sara Baqla and Julie Schlicte for data collection assistance, and the Sanders lab group and Brandon Bestelmeyer for input on previous drafts of this manuscript.

### Funding

Funding was provided by a National Science Foundation postdoctoral research fellowship award number 1401717 awarded to Israel Del Toro, and University of Texas El Paso and Jornada Long Term Ecological Research Station Research Experience for Undergraduate programs. The funders had no role in study design, data collection and analysis, decision to publish, or preparation of the manuscript.

### Grant Disclosures

The following grant information was disclosed by the authors:
National Science Foundation postdoctoral research fellowship: 1401717.
University of Texas El Paso.
Jornada Long Term Ecological Research Station Research Experience for Undergraduate programs.

### Competing Interests

The authors declare there are no competing interests.

### Author Contributions

- Israel Del Toro and Relena R. Ribbons conceived and designed the experiments, performed the experiments, analyzed the data, contributed reagents/materials/analysis tools, prepared figures and/or tables, authored or reviewed drafts of the paper, approved the final draft.

### Field Study Permissions

The following information was supplied relating to field study approvals (i.e., approving body and any reference numbers):

United States Forest Service provided field permits: FS2700-25.

### Data Availability

The raw data and R code are available in the Supplemental Files.

### Supplemental Information

Supplemental information for this article can be found online at http://dx.doi.org/10.7717/peerj.6686#supplemental-information.

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
