# Peer review of "Variation in ant-mediated seed dispersal along elevation gradients"

_PeerJ, doi:10.7717/peerj.6686_

## Round 0.1 · original submission · Major Revisions

The reviewers were quite thorough and raise a number of important topics for consideration during your revision. Many of these suggestions are related to possible ways to rephrase or structure areas of the manuscript to better describe the rationale and results, or to correct interpretation issues with the literature cited. In addition to these many important suggestions, please pay particular attention to questions regarding your GLMM raised by reviewers 1 and 3. Is your model overdispersed and is "time" a relevant fixed factor if the measurements across time are not independent of each other? Please address these issues and respond to the reviewers' suggestions in your revision.

Reviewer 1 ·

Basic reporting

The writing is ok, but often disjointed with paragraphs that contain many ideas with poor structuring. The authors repeatedly misinterpret and misreport the literature cited. The figures are ok, but not terribly impressive.

Experimental design

The authors suggest how they fill an identified knowledge gap, but these type of studies have been done many times, suggesting they need to engage in a more thorough literature review. Also, they overstate the climate change angle. A better approach would be to compare and contrast the low elevation myrmecochory driven by granivorous ants and the high-elevation myrmecochory driven by omnivorous ants – the authors need to investigate the differences in these two groups which is confused in their Discussion.

Validity of the findings

The study design seems fine, but the statistical analyses need refined; in particular, the authors use non-normal error distribution without testing for (and accounting for if present) overdispersion, which grossly overstates significance if present.

Additional comments

Lines 25-26 – I am not aware of myrmecochory (the dispersal of seeds by ants) structuring insect communities, unless the authors mean the ants themselves, which should be said rather than "insects."

Lines 26-27 – A strong statement that is not supported by the vast body of literature on ant-mediated seed dispersal.

Line 36 – Something missing here.

Lines 46-48 – A very general and open statement that does not lead into the next sentence well. I would save any climate change discussion until the Discussion as you measured spatial variation not temporal trends.

Lines 48-50 – Not true and not the correct citation for this statement. Zelikova et al. reported that other studies estimated 20-50%.

Line 57 – Ecosystem services generally mean those benefiting humans, not structuring vegetation community structure.

Lines 46-58 – what is the topic of this paragraph?

Lines 63-64 – again, overstated and not true, including citations included in this paper (e.g., Zelikova et al. who studied seed dispersal by elevation).

Lines 59-70 – again, what is the topic of this paragraph?

Line 71 – Zelikova et al. 2008 found just the opposite, ant community composition changed with elevation but myrmecochores did not.

Line 74 – you do not measure nor assess changes in climate; you are using a geographic gradient.

Lines 85-96 – How did the experimental presentation of seeds coincide with the natural seed set? The native plants would be expected to set seed at different points given the change in temperature regimes with elevation. How did the natural seed set of your study species coincide with your experimental presentations?

Line 108 – typo

Line 115 – You do not report any tests for overdisperion in the Poisson-distributed GLMM model. Overdispersion grossly inflates Type I error if present. Also, what is the random effect in the GLMM model?

Lines 116-117 – Why not just the sum of all seeds removed at each station?

Line 119 – Report ANODEV, and it appears you used lmer (assuming Gaussian error), which is an LMM not GLMM.

Lines 128-130 – Actually, based on your figures, it does not look like seed removal rates were dramatically different, except for one species, and the higher elevation ants were just slower.

Lines 132-134 – Why are you hypothesis testing time for seeds to be removed? Would you expect anything else?

Lines 153-156 – Wow, this is a complete misinterpretation of Warren and Giladi 2014 and Zelikova et al. 2008, who identified 'effective' dispersers as those that remove a lot of seeds intact and place in safe sites intact. Granivorous ants are a completely different type of seed dispersal and not what either paper addressed. Indeed, granivorous ants eat the seeds, which makes them less than effective seed dispersers.

I stopped here. The authors need to consider a very thorough revision including a review of the literature used.

Reviewer 2 ·

Basic reporting

Language used throughout: The language used is clear and easy to read and there are few grammatical errors.

Intro and background context and relevancy of literature: The introduction does a good job of explaining what myrmecohory is and how it may shape vegetation structure. The literature on myrmecochory and the plant structure associated with this relationship appears to be adequate. I do, however, feel the environmental gradient component, which is vital to the story, could be amplified a bit. For instance, Ln 62-64 suggestion that seed dispersal rates have not been studied along environmental gradients is notable. Adding on another sentence why this information is important to understand (e.g., vary with climate change? Loss of species? Etc.) will help the reader understand the potential impact of the study. If there are no studies exactly like this, including additional literature of how similar relationships are important along an elevational gradient could also be useful to bolster the importance of the study.

Structure of manuscript: Other than having a different reference format (see suggestions in "general comments" on Ln 217 onward) the structural layout of the manuscript follows Peer J guidelines.

Figure relevancy: Figures provided are relevant and effectively demonstrate the findings between elevation and species removal over time. Was the omission of the PCA of ant communities on purpose? If so, justification of why it is added to the statistical description and not the results is needed. If it was accidentally omitted providing a figure of the PCA could be useful for interpretation of the ant communities.

Raw data: The raw data provided are sufficient and helpful in understanding how the data were collected.

Experimental design

Relevancy of study within scope of journal and originality: This study falls within the scope of Peer J and appears is original in its questions, design, and findings.

Definition of research questions: Research objectives are defined so that the reader knows what the authors expect to find in their two primary predictions. If the authors want to keep the different questions defined as #1, and #2, I would find it a bit easier to follow if the two primary questions are clearly noted as questions, rather than predictions, and then referred back to in the methods, results, and discussion. Alternatively, it is fine to keep purpose of the study broad so that elevation and species are the two primary topics of investigation, however, if this is the format I would avoid using terms such as “in our second objective” as this leads the reader to think they missed the first (see "general comments" suggestion for line 98). It would also be helpful to maintain a consistency in stating “objectives” (e.g., Ln98) and “predictions” (e.g., Ln 130).

Finally, the introduction of the ant communities feels a bit misplaced. Why is this not a research question but is noted in the statistical methods, discussion, and r-code but not in the results? The decision to include this information or fully omit it would help continuity.

Rigorous nature of investigation: The investigation of using different species in addition to the elevational gradient helps fuel our understanding of how ants are moving around these seeds. I am comfortable with the level of rigor this study provides.

Detail of methods: The methods are generally easy to follow yet at this point don’t feel like they could be replicated with the information given. I would recommend more detail in the methods to help the reader easily understand how the two objectives are being addressed and implemented in the field. For instance, the statement of how the second objective will be tested but not the first big objective of rates of seed removal along elevational gradients isn’t totally clear (see specific methods suggestions in "general comments").

Validity of the findings

Rationale and benefit: Knowing the rate of seed removal by species and elevation will benefit the ecological community in understanding plant-insect interactions. It will also help the restoration ecology community as they often put down seeds as a treatment and do not know if seeds are removed from the site or if they just don’t germinate.

Stats and data robust and sound: Most of the statistics are stated in an unambiguous manner and the GLMMs fixed and random factors clearly defined. I would appreciate a bit more information on how the repeated measures ANOVA was conducted (see specific suggestions in "general comments") to better assess the accuracy of using this analysis.

Conclusions linked to study: The conclusions link to the study well and are sufficient in summarizing the big picture points the study identified.

Level of speculation: It doesn’t appear that the authors deviate too far outside their study topics in the discussion.

Additional comments

Minor comments with associated line numbers:

Ln 34-36: The closing paragraph of the abstract does not appear to be finished. …suggesting that temperature variation along the elevation gradient…? Please revise for clarity.

Ln 87: Please describe what a “seed depot” is and its intended purpose.

Ln 98: It is useful to remind the reader what your specific objectives are. I would recommend paraphrasing what your second objective is before jumping in to the individual species.

Ln 97-107: The information on the different plant species is very important and useful for the read, however, you don’t note how you distributed the different species seeds on the landscape or quantified their removal. Are the 10 deed depots used for both objective 1 and2? Please include this information.

Ln 118: Was the repeated measures ANOVA run in R too? More details in the R-code provided in the supplementary material and in the text are needed to accurately assess how this analysis was ran. Also, in Table 2 Site is included in the output but not noted that it is included in the model in the methods section.

Ln 136: include a space: and 18%

Ln 193: I like the reference to the phenology of sumac's seed drop and how it varies. One question I have wondered throughout: is there potential for temperature (or other climate variability) across the elevational gradients to guide the results? I don’t believe July is too early for the ants to be in full foraging mode, particularly in these SW systems, but could the difference in climate metrics across elevation gradients interact with the movement of the ants and their seed removal?

Ln 217: The format of references cited does not exactly follow peer J guidelines. For example the year is in parentheses and the volume number is in parentheses but not asked to be included in the references. See example reference:

Smith JL, Jones P, Wang X. 2004. Investigating ecological destruction in the Amazon. Journal of the Amazon Rainforest 112:368-374 DOI: 10.1234/amazon.15886.

Figure 1. “mean seed removal rates per bait station...” do you mean seed depot? If they are different please explain, if they are the same thing, I would stay consistent in your terms.

·

Basic reporting

The article is well written and cited, following the standard format of peer review publications. There are some citations that do not match the context for which they are given (listed below). This makes me worry that other citations are not representative, though I have not checked all of them and can only speak to the citations that reference me.

Zelikova et al 2008 (my paper) does not talk show that ant seed dispersal shapes plant communities, the Zelikova et al. 2011 paper does.

The claim in the discussion that the harvester ants have multiple traits which make them ants ideal dispersers cites Warren and Giladi 2014 and Zelikova et al. 2008 - neither of these papers are about harvester ants. So I would either cite appropriate harvester papers or mention the ant traits that are relevant (what are the traits that make them ideal seed dispersers?)

Experimental design

The research questions are not clearly stated, but rather indirectly stated by way of predictions. I would prefer a clearly stated question that links the factors that change along an elevational gradient linked to the process being studied (seed dispersal), rather than predictions not linked to process. The statement that ant communities are resilient to climate change (Stuble reference) is immediately followed by saying that less is known about that in a different ecosystem. "Less is known" is not a compelling enough reason - are there reasons to suspect that desert ecosystems would act fundamentally differently? If predictions are the way to tell this story, I would still suggest changing the framing.
The prediction that seed removal would be higher at lower elevations as a result of higher ant abundances and different ant communities relies on something that needs to be tested (higher ant abundances and different ant communities) but a prediction should stand on its on, not contingent on another experimental factor. And I am not sure we talk about seeds as having native elevation ranges but rather plant distribution that peaks at certain elevations.

The experimental work was done to high ethical standard and is not technically rigorous but does not need to be given the simplicity of the task (measuring seed dispersal by ants is not a highly technical endeavor, which is what makes it accessible and scalable). In terms of measurements, looking at 3 elevations is the minimum needed to establish an elevational gradient, but I could also argue that the 3 sites are different, regardless of elevation (no way to fix this, but really should have >3 sites for an elevational gradient study). Even the low elevation site in this study could actually be a mid-elevation site in another study and the high elevation site is not high enough to eliminate ant occurrence and actually have the capacity to test how ant abundance affects seed removals.

I would also say that we say "occurrence" and not "incidence" when referring to counting # of times we collect a taxonomic group.

In the GLM analysis, I am not sure treating time as a fixed effect is valid given the fact that the number of seeds removed is contingent upon time and # of seeds removed in hour 2 is not independent of how many seeds were removed at hour 1. I like the repeated measures framework instead and don't think the first GLM is necessary or adds anything to the analysis or outcomes. In fact, if comparing total # of seeds is of interest, can sum across time (since time is not part of the research questions or predictions). If oat is actually the "control", I would want to see removal rates compared to that control rather than all 4 species compared on what is essentially equal footing. Or another place to make that distinction is in the discussion.

Validity of the findings

The results are clear and unsurprising.
A few improvements can be made. Stating "In all analyses (global model and seed specific models), time influenced the number of seeds that the longer the depots are active, the more seeds are removed, ranging from 25% to 58% over the 48-hour time period" seems obvious and unnecessary.

I would want to see the actual direction of the trend following statements like "Our repeated measures analyses suggested a strong interaction between elevation and seed species (Table 2, p < 0.001)." - follow up with what the interaction actually was.

This might be journal standard, but I would italicize the genus names for the plant species in the results section.

In the discussion, the authors switch to talking about granivory rather than seed dispersal, which is an important distinction. If this is really a story about seed dispersal and not seed consumption, need to keep those terms straight.

Why is it meaningful that the harvester ants were observed removing seeds of Datura and Sumac, as well as the control Oat, but never Iris? Seems like there needs to be follow up or a hypothesis about Iris.

Given the heavy emphasis on "native" elevation ranges etc, I did not see enough supporting information or enough of a foundation in the introduction to link to the conclusions that basically link native plant ranges with ant abundances. For example, this sentence seems to be unsupported, at least by this study: "We highlight that for some species (Datura and Iris) their ant-mediated dispersal rate is highest in their native elevation range, which may suggest strong mutualistic links between ants and these plant species." If that's indeed the big take-away from this study, it needs to be explored and supported more.

---

## Round 0.2 · Major Revisions

After hearing from the expert reviewers, I am recommending another chance at major revisions for this manuscript. Both reviewers noted that efforts toward addressing comments from the prior round of reviews were minimal and largely cosmetics. In some cases, your rebuttal letter describes a change that was then not present in the manuscript. In this instance, I will give you the benefit of the doubt and assume that changes could have been lost among versions or that an older version was uploaded by mistake. Regardless of the cause, it is important that you return an accurate rebuttal letter that reflects the changes that were made. Also, recognize that the reviewers have willingly donated their time to offer thoughtful critiques and suggestions for improving this manuscript. In the case that you disagree with their comments and are choosing not to make a recommended change, a detailed explanation of why you disagree is required on a point by point basis so that I can make a decision as the editor.

Reviewer 2 ·

Basic reporting

Clear, unambiguous, professional English Language is used throughout.

Intro & background to show context and why the study is relevant. I, however, do feel that with the introduction changes that were already made the final paragraph could include directed research questions and the associated predictions. This would then allow the reader to know what the study was looking at what it expected to find. Then in the discussion efforts can be directed towards how the questions were answered and whether predictions were met, why or why not based on the current literature. This was brought up by a reviewer in the initial review but not fully addressed in the rebuttal.

Literature is well structured and relevant to the study that was conducted.

Figures are relevant and visually augment the text. The figures could, however, be changed so that there are no graphical lines (horizontal or vertical) within the main frame of the figure. The removal of any grid lines will increase the visibility of the general observed trends over time and the differences between elevations.

Experimental design

This primary research is original and demonstrates an interesting insect plant relationship across an elevational gradient.

Addresses an identified knowledge gap which appears to meet a high ethical standard when disseminating the information. The methods have been improved to include greater detail on the methods and the information available in the case someone would want to replicate their study.

Validity of the findings

Data appear robust, statistically sound and with the adjustments to the statistical efforts (e.g., testing for overdispersion) appear to be sound.

Is any other insect moving these seeds around? Acknowledgment of other ways the seeds may have been removed from the seed depot should be acknowledged. Alternatively, if only ants can remove the seeds this should be directly stated other than just ‘larger granivores’.

Discussion and conclusions are fine.

Additional comments

General comments:
Please be sure to address the reviewer comments/suggestions completely and be directed in the responses. A few times suggestions were made by a reviewer and it was stated that the changes were made but they were not necessarily fully addressed. It is totally fine to politely disagree with a reviewer with explanation as to why, but if changes are acknowledged as made, please be directed in how they are made. For instance, the statement "highlighted the broader significance of the study in the discussion" was made as a response to a suggestion about building the link to the broader context in the introduction, but then when looking over the discussion, it is not clear where the change(s) were made.

Line comments:
Ln 58: With the change in text “sky Islands” now seems to come out of nowhere. Including one sentence linking the two paragraphs would fix this. For example, because the last sentence ends with a focus on dispersal the beginning of the next sentence could read something like: “While seed dispersal can play a key role in vegetation communities, its impacts can be influenced by elevational gradients…” or something of the sort.

Ln 74: A bit more information is needed to help the reader. Perhaps, as an example, you could start with: “To better understand myrmecochory across elevation gradients and species specific ant-plant mutualism, we documented….”

Ln 92: “(detailed below)” still isn’t totally clear. Lines 107-110 appear to be about the pitfall traps, not seed depots. Unless I am misunderstanding, these are not one and the same. Please see initial review comments as I don't believe this was fully addressed.

Ln 120: I like the inclusion of the over dispersion text/explanation.

Ln 188-191: Isn’t this low elevation observation of ant communities having a highly active and seed dispersing species just a repeat of how harvester ants, and the other species, are active dispersers in the previous paragraph?

Ln 198: It is nice to see when “future work” is suggested, but a bit more clarity, and directed detail on how this is different than this study, is needed when the statement “the network of ants interacting with various seed species being dispersed along environmental gradients” is made.

·

Basic reporting

The article meets the basic standards of being written in clear professional English, with references and background provided. In general, this research is not novel, at least the way it is presented here.

Experimental design

This is primary research. The research questions, however, are not well linked to the background and context provided. The study is carried out well and there are no huge issues with methods, which are simple and easy to replicate. The issues are related to the reasons for the study and what the outcomes tell us. Given the fact that the authors essentially performed a preference study with ants and seeds across an elevation gradient limits the study's applicability, especially because the results are not linked to the study's framing. Specific comments below:

Validity of the findings

See below and appended

Additional comments

This manuscript presents a seed removal studies conducted across an elevation gradient. These studies have been done a lot, with seeds dispersed by ants and by other dispersal agents. As presented in this manuscript, this study is not original and does not add any new information to our understanding of how systems function, what the impact of this kind of seed dispersal is on the resultant plant communities, and it misses the mark on doing the relevant analyses framed in the research questions. For example, no meaningful analyses of the influence of ant community compositional shifts across the elevation gradient are conducted. Elevation does not in and of itself move seeds so biological processes needed to be invoked. Since I did not see meaningful progress from the last iteration of this manuscript and the edits were cosmetic and not substantial, I recommend to reject this manuscript from publication. I provide more detailed comments throughout the manuscript.

---

## Round 0.3 · Minor Revisions

I have now received feedback from one of the initial reviewers on your manuscript. Given their comments, I feel that a round of minor revisions to address a few lingering topics is needed before publication. Please be thorough in dealing with the reviewer comments and add clarity to any discipline-specific definitions or methodology.

Reviewer 2 ·

Basic reporting

No comment, meets journal standards.

Experimental design

After changes to previous versions appears to meet journal and scientific technical and ethical standards.

Validity of the findings

No comment, meets general standards.

Additional comments

In reviewing the resubmitted manuscript by Del Toro and Ribbons, Variation in ant-mediated seed dispersal along elevation gradients, my first comment is regarding the rebuttal letter. The rebuttal letter itself feels incomplete. The authors state that they have conducted line by line edits, and while the most recent review did not contain as many line by line edits suggestions as the previous versions, it did contain some directed instructions to provide greater clarity and depth to suggested changes, from both myself and the other reviewer. With such a short rebuttal letter, the burden is now put on the reviewer to hunt and peck where the changes were made, whether specific or broad, and decide not only if they have found the right place, but if they were then sufficient. For future manuscripts (or perhaps this one) I strongly recommend directly answering the reviewers comments in a clear and concise manner in both the rebuttal letter as well as the main text document. This then allows the authors, reviewers, and editor to all be on the same page and move the dialog as well as scientific efforts forward.

An example of where I believe the changes were made but had to go back over my previous review to be sure, is the seed depots. In looking back over my previous two requests I ask that the difference of seed depots and pitfall traps are clearly differentiated. While each time there is an incremental change, the definition, or explanation, of what a seed depot is, for the broad Peer J audience that may not know, is never included. I understand this is may feel like a minor request to the authors, but it leads me to wonder what else hasn’t been directly addressed.

That said, I do appreciate the authors changes, the effort put in, and feel the manuscript only requires minor revisions before being sufficient for publication:

Minor edits (using the rebuttal text line numbers):
Ln 85-87: A brief explanation of why species-specific responses in addition to the heterogeneous environment may occur would help this sentence.
Ln 130: How close were these to your seed depots?
Ln 206-208: What was a generality in the ant-Iris interactions and how could it relate to your work?

---

## Round 0.4 · accepted · Accept

Thank you for attending to the comments raised by the reviewers throughout the process. In reading through your manuscript, I caught a small typo: on line 122 there is no space between "a" and "16". PeerJ does not offer copy editing service so be sure to carefully read your final files and proofs as the process moves forward to ensure that your articles content will match expectations. Congrats on the paper!

#